# Recent Developments in Human Papillomavirus (HPV) Vaccinology

**DOI:** 10.3390/v15071440

**Published:** 2023-06-26

**Authors:** Anna-Lise Williamson

**Affiliations:** Institute of Infectious Disease and Molecular Medicine/SAMRC Gynaecological Cancer Research Centre/Division of Medical Virology, Department of Pathology, University of Cape Town, Cape Town 7925, South Africa; anna-lise.williamson@uct.ac.za; Tel.: +27-834628798

**Keywords:** papillomavirus, HPV vaccine, cervical cancer, anal cancer

## Abstract

Human papillomavirus (HPV) is causally associated with 5% of cancers, including cancers of the cervix, penis, vulva, vagina, anus and oropharynx. The most carcinogenic HPV is HPV-16, which dominates the types causing cancer. There is also sufficient evidence that HPV types 18, 31, 33, 35, 39, 45, 51, 52, 56, 58 and 59 cause cervical cancer. The L1 protein, which, when assembled into virus-like particles, induces HPV-type-specific neutralising antibodies, forms the basis of all commercial HPV vaccines. There are six licensed prophylactic HPV vaccines: three bivalent, two quadrivalent and one nonavalent vaccine. The bivalent vaccines protect from HPV types 16 and 18, which are associated with more than 70% of cervical cancers. Prophylactic vaccination targets children before sexual debut, but there are now catch-up campaigns, which have also been shown to be beneficial in reducing HPV infection and disease. HPV vaccination of adults after treatment for cervical lesions or recurrent respiratory papillomatosis has impacted recurrence. Gender-neutral vaccination will improve herd immunity and prevent infection in men and women. HPV vaccines are immunogenic in people living with HIV, but more research is needed on the long-term impact of vaccination and to determine whether further boosters are required.

## 1. Cancers Associated with Human Papillomavirus

Human papillomavirus (HPV) is causally associated with 5% of cancers, including cancers of the cervix, penis, vulva, vagina, anus and oropharynx [1,2,3]. There is also evidence of HPV’s role in conjunctival cancers [4]. The most important HPV-associated cancer is that of the cervix (refer to Figure 1); over 95% of cervical cancers are associated with HPV, and in 2018, there were about 570,000 new cases and 311,000 deaths globally [5]. The burden of cervical disease is unevenly distributed, with more than 85% of cases and deaths occurring in low-income and middle-income countries [6]. These differences are primarily due to poor cervical cancer screening and treatment programmes [7]. HPV is associated with up to 70% of oropharyngeal cancers, and oral squamous cell carcinomas (OSCCs) in Europe and North America continue to rise [8]. In the United States, oropharyngeal cancers now outnumber cervical cancer cases and are the number one HPV-related cancer [9]. While the role of HPV in oesophageal cancers remains uncertain, HPV can be detected in a subgroup of these cancers [10,11]. A study conducted in South Africa demonstrated that oesophageal cancer was statistically significantly associated with increased anti-HPV-16 IgG antibody levels [12]. It was estimated that, globally, there were 30,416 cases of squamous cell carcinoma of the anus caused by HPV on an annual basis. Two-thirds of these cases occurred in women [13].

## 2. Structure of Human Papillomavirus

Papillomavirus particles are around 60 nm in diameter and comprise 72 pentameric capsomeres enclosing a circular, double-stranded DNA genome [14]. HPV genomes encode a regulatory region, two structural proteins (L1 and L2) and a number of early proteins (E1-7). The early proteins play a role in the virus’s replication as well as in carcinogenesis in high-risk HPVs. L1 is the major capsid protein, and L2 is a minor structural protein making up the capsomere. The L1 protein self-assembles into virus-like particles, forming the basis of the current HPV vaccines [15,16,17].

HPVs belong to the family *Papillomaviridae* and subfamily *Firstpapillomavirinae.* Most HPVs fall into the genera *Alphapapillomavirus*, *Betapapillomavirus* and *Gammapapillomavirus.* The classification of HPV is based on the sequence of the L1 gene [18]. The different subfamilies share less than 45% L1 sequence identity, and different genera have less than 60% sequence identity compared to other genera [19]. Within the genera, the classification of HPVs is further classified into types or species based on the sequence of the L1 gene. HPV types have an L1 sequence divergence of at least 10% from that of any other PV type [18]; those with 2–10% divergence are known as subtypes, and those with less than 2% are called variants [20]. The naming of HPV types is the responsibility of the International HPV Reference Centre (Hpvcenter.se). To describe a novel HPV type, the DNA must be cloned and sequenced, and the clone and sequence must be submitted to the International HPV Reference Centre, where it will be assigned a unique HPV type number if it is novel. There are 221 official HPV types. However, there are also databases that have sequences of HPV from viruses that have not been cloned, in which a further 592 putative novel HPV types exist, bring the number HPV types to over 800 [21].

## 3. HPV Types Associated with Cancer and Genital Warts

Specific types of HPV are classified as carcinogenic to humans (Group 1). The most carcinogenic HPV is HPV-16, which dominates the types causing cancer. There is also sufficient evidence that HPV types 18, 31, 33, 35, 39, 45, 51, 52, 56, 58 and 59 cause cervical cancer. The HPV types 26, 53, 66, 67, 70, 73, 82, 30, 34, 68, 69, 85 and 97 had limited evidence of causing cervical cancer [22]. HPV-16 and 18 are responsible for about 70% of cancers worldwide [23,24,25,26]. The most prevalent HPV types found in cancers have informed the HPV types detected in screening and vaccination. GARDASIL9^®^ has VLPs from HPV types 6, 11, 31, 33, 45, 52 and 58, which should protect from 90% of the HPVs causing cervical cancer, while the bivalent and quadrivalent vaccines should protect from 70% of infections [27]. In Africa, HPV-35 has been found in up to 10% of cancers [28,29], and this type is not present in the current vaccines.

Over 90% of genital warts are caused by HPV-6 and HPV-11 [30]. The nonavalent and quadrivalent vaccines protect from genital warts as well as from cancer-causing HPV types.

## 4. HPV Vaccines

The HPV L1 protein, which, when assembled into virus-like particles (VLPs), induces HPV-type-specific neutralising antibodies, forms the basis of all commercial HPV vaccines. The first proof of concept that VLPs could protect from infection came from animal models. The canine oral papillomavirus (COPV) L1 gene was produced in a baculovirus expression system and injected into beagles. The vaccinated animals were resistant to experimental challenges with COPV. In addition, passively infused serum immunoglobulins also protected from infection, confirming that the protection was antibody mediated [31]. Further confirmation of L1 VLP-based vaccines being effective prophylactic vaccines was provided by challenge models with bovine papillomavirus [32] and cottontail rabbit papillomavirus [33]. These results and proof that HPV L1 could assemble into VLPs [15,16,17] provided proof of concept justifying the development of HPV vaccines.

Currently, there are six licensed prophylactic HPV vaccines: three bivalent, two quadrivalent and one nonavalent vaccine (refer to Table 1). The first efficacy trials on HPV prophylactic vaccines were on GARDASIL^®^ and Cervarix^®^. GARDASIL^®^ is a quadrivalent vaccine carrying HPV-16, HPV-18, HPV-6 and HPV-11 VLPs. Cervarix^®^ is a bivalent vaccine with HPV16 and HPV-18 VLPs. Both vaccines gave excellent efficacy in protecting from incident infection with the vaccine HPV types as well as cervical intraepithelial neoplasia grade 3 and had a good safety profile [34,35,36,37,38]. Protection from vaginal/vulvar lesions and genital warts was observed in the GARDASIL^®^ trials [39,40]. Less information is available on the newer HPV vaccines. Two new Chinese vaccines were tested in a randomised, blinded, non-inferiority phase III trial to determine the efficacy of their novel four- and nine-valent HPV vaccines (4vHPV, HPV 6/11/16/18; 9vHPV, HPV 6/11/16/18/31/33/45/52/58) and were proved to not be inferior to GARDASIL^®^ (4vHPV, HPV 6/11/16/18) in terms of immunogenicity and safety [41]. GARDASIL9^®^ is indicated not only for anogenital lesions in men and women up to the age of 45, but recently, the FDA also included oropharyngeal and other head and neck cancers caused by HPV types 16, 18, 31, 33, 45, 52 and 58 as an indication for this vaccine [42].

HPV vaccination induces high levels of HPV-type-specific antibodies, much higher than the titres induced during natural infection. The immunisation of children <16 years gives significantly higher antibody titres than in older children and adults [43]. The drop-off with time in titres after GARDASIL^®^ or GARDASIL9^®^ vaccination differs across the HPV types in the vaccine, with HPV-18 and 45 antibodies dropping faster than other HPV-type antibodies. Despite the drop in titres, there still appears to be protection from infection [44]. Cervarix^®^ induced higher antibody titres than GARDASIL^®^. In a study of neutralising antibodies in women who had been vaccinated with either GARDASIL^®^ or Cervarix^®^, 15% of the 339 GARDASIL^®^ recipients had no detectable HPV-18-neutralising antibodies 2–12 years after vaccination. In contrast, all 342 Cervarix^®^ recipients had HPV-18-neutralising antibodies. The HPV-16 geometric mean titres halved after 5–7 years compared to 2–4 years after vaccination with GARDASIL^®^ [45,46]. The higher antibody titres induced by Cervarix^®^ also resulted in more cross-protection from other HPV types related to those in the vaccine [46]. Thus far, there are no correlates of protection for HPV vaccines that state the titre of antibody needed to protect from infection as the minimum antibody titre required to protect from infection is unknown. The antibody tests are also not standardised, which makes comparisons between studies challenging [47].

Ongoing monitoring of HPV vaccination worldwide continues to confirm the efficacy of the vaccines as well as the safety profile of the vaccines [48,49,50]. In the USA, HPV prevalence was compared in the vaccination era to that observed in the pre-vaccination period. The impact of GARDASIL^®^ vaccination among sexually experienced females was 90% among vaccinated females and 74% among unvaccinated females, indicating the effect of herd immunity in the unvaccinated people. No impact was seen on non-vaccine HPV types [51]. In Spain, a similar reduction in the prevalence of vaccines related to HPV types and a significant decrease in premalignant cervical lesions was observed [52]. However, one study reported that, although the vaccine types were controlled, the percentages of HPV-31, HPV-52 and HPV-45 had increased [50].

A substantial decrease in genital warts was seen on gender-neutral HPV vaccination introduction in schools [53]. Juvenile-onset recurrent respiratory papillomatosis (JORRP), thought to be acquired from the mother during birth, appears to be declining in the USA. The decline is believed to be due to HPV vaccination, and once vaccine coverage is high enough, this disease should disappear [54].

**Table 1 viruses-15-01440-t001:** Characteristics of six HPV vaccines.

Vaccine Name	Valency VLP Types	Manufacturer and Licensure Date or WHO Prequalification Date	Adjuvant	Expression System	Manufacturers’ Schedules
GARDASIL^®^ *	Quadrivalent, HPV types 6, 11, 16 and 18	Merck & Co., 2006	Amorphous aluminium hydroxy phosphate sulphate 225 μg	Yeast, *Saccharomyces cerevisiae* expressing L1	GARDASIL^®^ is licensed for girls and boys aged 9–13 years as a 2-dose schedule (6 months apart). From age 14, a 3-dose schedule should be given (at 0, 1–2 and 4–6 months).
Cervarix^®^ *	Bivalent, HPV-16, HPV-18	GlaxoSmithKline, 2007	0.5 mg aluminium hydroxide and 50 μg 3-0-desacyl-4′monophosphoryl lipid A	Insect cell line, recombinant baculovirus encoding L1	Cervarix^®^ is licensed for girls and boys aged 9–14 years as a 2-dose schedule (5–13 months apart). If the recipient’s age at the time of the first dose is ≥15 years, three doses should be given (at 0, 1–2.5 months and 5–12 months)
GARDASIL9^®^ *	Nonavalent, HPV types 6, 11, 31, 33, 45, 52 and 58	Merck & Co., 2014	Amorphous aluminium hydroxy phosphate sulphate 500 μg	Yeast, *Saccharomyces cerevisiae* expressing L1	GARDASIL9^®^ is licensed for girls and boys aged 9–14 years as a 2-dose schedule (5–13 months apart). From age 15, a 3-dose schedule should be followed (at 0, 1–2 and 4–6 months).
Cecolin^®^ *	Bivalent, HPV-16, HPV-18	Xiamen, Innovax Biotechnology, 2020	Aluminium hydroxide 208 μg	Bacteria, *Escherichia coli* expressing L1	Cecolin is licensed for girls aged 9–14 years as a 2-dose schedule (6 months apart). From age 15, a 3-dose schedule is indicated (at 0, 1–2 months and 5–8 months).
Walvax recombinant HPV vaccines—WalrinvaxV	Bivalent, HPV-16, HPV-18	Shanghai Zerun Biotechnology (a subsidiary of Walwax Biotechnology), 2022	Aluminium phosphate	Yeast, *Pichia pastoris* expressing L1	Walrinvax is licensed for girls aged 9–14 years as a 2-dose schedule (6 months apart, with a minimum interval of 5 months). From age 15, a 3-dose schedule is indicated (at 0, 2–3 and 6–7 months).
Cervavac^®^ **	Quadrivalent, HPV 6, 11, 16 and 18	Serum Institute of India, 2022	Aluminium based		Cervavac is licensed for girls and boys aged 9–14 years, as a 2-dose schedule (6 months apart). From age 15, a 3-dose schedule should be given (at 0, 2 and 6 months)

References [48,55,56], * prequalification by WHO, ** launched in India.

## 5. HPV Vaccine Implementation Globally

In May 2018, the Director General of the World Health Organisation (WHO), Dr. Tedros Ghebreyesus, called for action towards the elimination of cervical cancer as a public health problem with a 2030 target of 90% of girls to be fully vaccinated with an HPV vaccine by 15 years of age [57], which would reduce the incidence of cervical cancer by 42% by the year 2045 [58]. The world is not on target to meet this goal. By late November 2022, 125 countries (64%) had introduced the HPV vaccine in their national immunisation programme for girls, and 47 countries (24%) had included boys [56] (refer to Figure 2). The COVID-19 pandemic has had a significant impact on HPV vaccination in LMICs. An effort is needed to increase global investment in HPV vaccination by adopting a school-based delivery. The proposed one-dose campaign will accelerate coverage [59]. It is of concern that many of the countries with the highest incidence of cervical cancer (Figure 1) have not included the HPV vaccine in their national immunisation programmes (Figure 2). These countries would benefit more from the nonavalent vaccines than from bivalent or quadrivalent vaccines since the nonavalent vaccine will give broader protection. LMICs have introduced bivalent or quadrivalent vaccines into their national immunisation programmes. In contrast, the nonavalent vaccine is used mainly in high-income countries with good cervical cancer prevention and treatment programmes.

## 6. Evidence for Reduction of HPV Vaccine Triple Dose to Two- or One-Dose Regimens

Implementing three-dose vaccine campaigns in young girls has been challenging, so reducing the doses to two or one would positively impact vaccination campaigns. The manufacturers now recommend two doses for young girls and boys but a three-dose schedule for older boys and girls (refer to Table 1). A number of the original HPV vaccine efficacy trials had participants who received only one or two doses of vaccines instead of three. Post hoc analyses of these trials provided evidence of the reduced doses’ efficacy on immunogenicity and HPV protection. In a large trial in Costa Rica, the efficacy of Cervarix^®^ against incident HPV-16/18 infections for three doses was 77.0% (95% CI 74.7–79.1); for two doses, it was 76.0% (62.0–85.3) and for one dose, was 85.7% (70.7–93.7). There was a difference as to when the second dose was received, with vaccine efficacy against incident HPV-16/18 infection for two-dose vaccinated women who received their second dose at one month was 75.3% (54.2–87.5) and 82.6% (42.3–96.1) for those who received the second dose at six months [60]. In this cohort, the antibody titres in vaccinated participants were much higher than those observed in natural infection but had lower titres with fewer vaccine doses [61]. Since a correlate of protection has not been established for HPV vaccines, it is unknown what impact this will have on protection over time. In a study in India, at the ten-year follow up, it was reported that a single dose of the GARDASIL^®^ vaccine provided similar protection against persistent infection from HPV-16 and 18 to that provided by two or three doses [62].

A randomised, multicentre, double-blind, controlled trial of single-dose GARDASIL9^®^ or Cervarix^®^ HPV vaccination compared to meningococcal immunisation was performed in Kenya (women aged 15–20 years). One dose efficiently prevented incident persistent oncogenic HPV infection, similar to multidose regimens as tested over 18 months [63]. Several trials, planned or still in progress, will generate data to inform the implementation of the single-dose vaccination [64].

There is currently a lack of evidence for the effectiveness of one dose of HPV vaccine in boys.

## 7. World Health Organisation Recommendations on HPV Vaccination

In 2022, the World Health Organisation (WHO) Strategic Advisory Group of Experts on Immunisation (SAGE) concluded that evidence supported that a single-dose HPV vaccine protects against HPV at a level comparable to two-dose schedules [65].

The WHO now recommends the following:A one- or two-dose schedule for girls aged 9–14 years.A one- or two-dose schedule for girls and women aged 15–20 years.Two doses with a 6-month interval for women older than 21 years.

People living with HIV and those who are immunocompromised should receive three doses. If this is not possible, then at least two doses, as there are knowledge gaps in this community on reduced vaccine doses [56,65].

## 8. Catch-Up HPV Vaccination

National HPV immunisation is usually performed before sexual debut in children before age 15 and often targets 9–10 year olds. One strategy to increase vaccine coverage has been catch-up campaigns to increase coverage. The justification is that even if people are HPV positive, they may not be infected with all the HPV types in the vaccine; delayed immunisation will protect them from those types. There is also the possibility of reinfection with HPV types, as natural HPV infection does not protect from reinfection or reactivation of latent infections [66,67]. In England, women were vaccinated aged 24–25 and were compared with unvaccinated women aged 26–29. There was a decrease in the detection of HPV-16 and 18 (3 to 1% (*p* < 0.001)) and CIN2 (6 to 3% (*p* < 0.001)), confirming the effectiveness of the catch-up campaign [68]. A 19% reduction in CIN2+ rates was seen in a catch-up vaccination study of women aged 18–26 years in the USA [69]. In Italy, catch-up HPV vaccination (up to age 25) considerably reduced the risk of all cervical abnormalities and increased herd immunity [70]. Catch-up vaccination has been proved to have benefits and, in some populations, it is cost effective [71,72,73].

## 9. HPV Vaccination of People with HPV-Associated Disease

While the current vaccines are prophylactic, they have been found to have benefits in patients being treated for HPV-associated lesions, including cervical intraepithelial neoplasia (CIN), genital warts, anal neoplasia and recurrent respiratory papillomatosis (RRP). The mechanism of inhibition of further disease is thought to prevent further spread of the HPV. In some cases, differences in efficacy were observed when vaccination was carried out after treatment compared to vaccination before treatment. A systematic review was conducted of sixteen studies of women with cervical dysplasia who received the peri-operative HPV vaccine. Recurrences of CIN 1, 2 and 3 were significantly lower in the vaccinated than in the unvaccinated group [74]. A meta-analysis of 22 articles reporting on HPV-vaccinated women undergoing local treatment for CIN reported evidence that HPV vaccination reduced the risk of recurrence of CIN when related to HPV-16 or HPV-18 [75]. In women with CIN, HPV vaccination post-treatment was associated with a significantly reduced risk of CIN2+ recurrence [76]. Further, randomised controlled trials are needed to confirm this result and inform patient management.

RRP is a relatively rare disease with papillomavirus in the respiratory tract caused by HPV types 6 and 11 [77]. Vaccination of patients with RRP resulted in a significant increase in time between surgical procedures with the implication that it is beneficial [78]. In a non-randomised trial, patients with newly diagnosed or established RRP were vaccinated, and both groups showed a reduced frequency of RRP and a reduced disease burden [79]. A meta-analysis of HPV vaccination in RRP cases alongside surgery indicated that vaccination is a beneficial adjunct therapy alongside surgery [80]. Treatment with GARDASIL9^®^ has positively affected five cases of recalcitrant genital warts [81]. Further trials are needed to determine whether this treatment should be recommended as a standard of care.

## 10. The Case for Vaccinating Boys and Men

There are two aspects to be considered in gender-neutral vaccination. The first is the contribution to herd immunity, and the second is preventing HPV-associated disease in men. One of the issues in recent history has been the shortage of HPV vaccines and, therefore, the prioritisation of vaccinating girls over boys [82,83]. The ethics of introducing gender-neutral vaccination in a time of vaccine shortage was a subject of debate because it meant that girls in high-prevalence cervical cancer areas could not obtain vaccines [82,84]. However, there are now more vaccine manufacturers [56] and, hopefully, the vaccine shortage is in the past and less expensive vaccines will become available. In the quest to prevent cervical cancer, gender-neutral vaccination would improve the elimination of those HPV types targeted by the vaccines [85].

HPV is sexually transmitted; therefore, protecting boys from HPV infection improves herd immunity and further protects their partners. The latter is particularly true where the vaccine coverage may not be optimal [86]. Although HPV vaccination of females has been shown to protect men from HPV in their cohort [87], this assumes that sexual contact will be within this cohort. In some countries, a key driver of HIV transmission is the age gap between the girl and her partner and sexual partnering outside of the age cohort [88,89].

Genital warts are the most prevalent HPV-associated disease in men and are of substantial cost to the healthcare systems [90,91]. Oropharyngeal and anogenital cancers in men are often diagnosed late because there are no routine screening methods, and management is still being researched [92,93,94]. In terms of preventing HPV disease in men, gender-neutral HPV vaccination has been found to be cost effective in many countries—although these tend to be high-income countries [71,95]. As more vaccines become available, vaccines will likely be less expensive, and more countries can then justify gender-neutral vaccination based on cost.

## 11. HPV Vaccination in People Living with HIV

Men and women living with HIV are more at risk of persistent HPV infections and associated cancers, including anal and cervical cancer. They have a higher prevalence of HPV infections, higher HPV viral load and are more likely to be infected with multiple HPV types [96,97]. HPV is more persistent in HIV-infected people, so older age groups have a higher prevalence of HPV than HIV-negative people [98]. Women with HPV are more likely to be infected with HIV than HPV-negative women [99,100].

Women with HIV who contract HPV are more likely to have anogenital disease than HIV-negative women [101,102]. Cervical cancer is an AIDS-defining illness [103], and 85% of women with cervical cancer live in Sub-Saharan Africa, where 24.9% of cervical cancers were diagnosed in HIV-positive women compared with 1.3% in the rest of the world [104,105]. HIV-positive women develop cervical cancer at a significantly younger age than HIV-negative women, so they must be screened at a younger age [103]. After treatment, cervical disease is more likely to recur in HIV-positive women than in HIV-negative women [106]. HIV-positive men and women are at a higher risk of squamous carcinoma of the anus. This cancer has been reported at a higher incidence in women but is also significant in men. In particular, men who have sex with men are at a higher risk of anal cancer compared to those who have sex with women [107].

The high prevalence of HPV-related disease in HIV-positive individuals emphasises the importance of HPV vaccination in this group of people. The WHO recommends at least two but preferably three vaccine doses in HIV-positive individuals [56]. In a meta-analysis of HPV vaccines in people living with HIV, Staadegaard et al. (2022) report that while seropositivity remained high after three vaccine doses, there was some decline over time, especially against HPV-18 and for GARDASIL^®^. There are no extensive studies on the efficacy of HPV vaccines in HIV-positive people. The significance of these declines is unknown, so this cohort needs to be monitored for HPV infection and disease to determine when and whether they need vaccine boosters. Catch-up vaccination on immunosuppressed women aged 18–26 did not significantly decrease CIN2 [69].

Of concern is the effect of implementing a one-dose vaccination in a population with high HIV prevalence. South Africa has the largest number of people living with HIV (estimated 7.9 million), with young women carrying the burden of HIV incidence [89]. HPV knowledge gaps include the impact of HPV vaccination before HIV infection and whether HIV will impact further vaccine efficacy. The effect of changing to one dose from two-dose vaccination also needs investigation. There is also a knowledge gap on the impact of catch-up vaccination campaigns or vaccination after treatment for HPV-associated lesions in HIV-positive people.

## 12. Conclusions

HPV vaccination has been proved effective in preventing specific types of HPV infection and significantly impacts HPV-associated disease, particularly cervical cancer. It is a priority that every effort must be made to vaccinate young people against HPV, particularly in those regions where cervical screening programmes are inadequate. Catch-up vaccination campaigns would protect a significant number of extra people from disease. Where feasible, vaccines should also protect from genital warts as they cause a considerable disease burden. Boys should be vaccinated to protect their partners from HPV infection and themselves from HPV disease. Finally, more research is needed on how to safeguard HIV-positive individuals from HPV-associated disease and what the role of HPV vaccination should be.

## Figures and Tables

**Figure 1 viruses-15-01440-f001:**
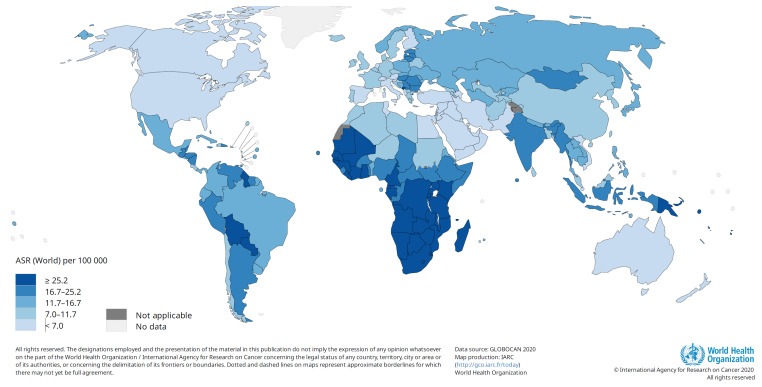
Estimated age-standardised incidence rates: cervical cancer (GLOBOCAN 2020). IARC/WHO is acknowledged for permission to use the figure from http://gco.iarc.fr/today accessed on 26 June 2023.

**Figure 2 viruses-15-01440-f002:**
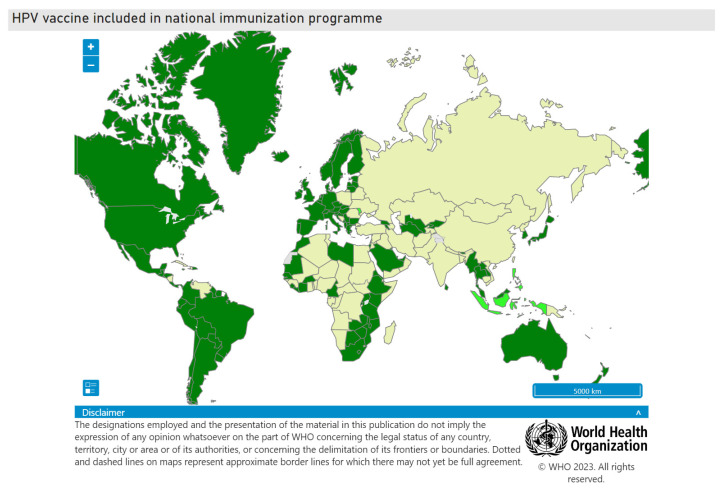
The status of HPV vaccine introduction in the world with the countries in dark green having introduced HPV vaccination. WHO is thanked for permission to publish this figure from https://www.who.int/teams/immunization-vaccines-and-biologicals/diseases/human-papillomavirus-vaccines-(HPV)/hpv-clearing-house/hpv-dashboard accessed on 26 June 2023.

## Data Availability

No new data were created.

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
