# Peer review of "Recent Developments in Human Papillomavirus (HPV) Vaccinology"

_viruses, 2023, doi:10.3390/v15071440_

Round 1

Reviewer 1 Report

Review

Recent Developments in HPV Vaccinology

Anna-Lise Williamson

The manuscript contains an elaborate literature review on a very topical topic such as HPV vaccination. The paper appears clear, complete and up-to-date. I was surprised that the author do not refer to the fact that since 2020 the FDA has opened the use of the nonovalent vaccine also for males aged 12 to 45 with the specific indication to prevent oral cancer.

Author Response

Comment:  I was surprised that the author do not refer to the fact that since 2020 the FDA has opened the use of the nonovalent vaccine also for males aged 12 to 45 with the specific indication to prevent oral cancer.

The article has been updated to include this information:

 GARDASIL9® is indicated not only for anogenital lesions in men and women up to the age of 45, but recently, the FDA included oropharyngeal and other head and neck cancers caused by HPV types 16, 18, 31, 33, 45, 52 and 58 as an indication for this vaccine [42].

Reviewer 2 Report

Well written manuscript.

A few lines on the peak antibody titres achieved and sustainability of the anti HPV antibodies after vaccination, vis a vis the age of vaccination, can be incorporated.

The methodology for detection of antibodies may also be mentioned.

Author Response

Comment:

A few lines on the peak antibody titres achieved and sustainability of the anti HPV antibodies after vaccination, vis a vis the age of vaccination, can be incorporated.

The methodology for the detection of antibodies may also be mentioned.

Response: I added this paragraph

HPV vaccination induces high levels of HPV-type-specific antibodies, much higher than the titres induced during natural infection. Immunisation of children (<16 years) gives higher significantly higher antibody titres than in older children and adults [43]. The drop-off with time in titres after GARDASIL® or GARDASIL9® vaccination differs between the HPV types in the vaccine, with HPV-18 and 45 antibodies dropping faster than other HPV-type antibodies. Despite the drop in titres, there still appears to be protection from infection [44]. Cervarix® induced higher antibody titres than GARDASIL® .  In a study of neutralizing antibodies in women who had been vaccinated with either GARDASIL® or Cervarix®, 15% of the 339 GARDASIL® recipients had no detectable HPV-18 neutralising antibodies 2-12 years after vaccination. In contrast, all 342 Cervarix® recipients had HPV-18 neutralising antibodies.  The HPV-16 geometric mean titres halved after 5 – 7 years compared to 2 – 4 years after vaccination with GARDASIL® [45,46].  The higher antibody titres induced by Cervarix® also resulted in more cross-protection from other HPV types related to those in the vaccine [46].  As yet, there are no correlates of protection for HPV vaccines which state the titre of antibody needed to protect from infection as the minimum antibody titre required to protect from infection is unknown. Antibody tests are also not standardised, which makes comparisons between studies challenging [47].